# Barriers and facilitators to parents' engagement with and perceived impact of a childhood obesity app: A mixed-methods study

Madison Milne-Ives[1,2], Em Rahman[3], Hannah Bradwell[4], Rebecca Baines[5], Timothy Boey[6], Alison Potter[3], Wendy Lawrence[7], Michelle Helena van Velthoven[8], Edward Meinert[1,2,9,10]*

1 Translational and Clinical Research Institute, Faculty of Medical Sciences, Newcastle University, Newcastle upon Tyne, United Kingdom, 2 Centre for Health Technology, School of Nursing and Midwifery, University of Plymouth, Plymouth United Kingdom, 3 South East School of Public Health, Workforce Training and Education Directorate, NHS England, United Kingdom, 4 School of Nursing and Midwifery, Faculty of Health, University of Plymouth, Plymouth, United Kingdom, 5 Peninsula Medical School, Faculty of Health, University of Plymouth, Plymouth, United Kingdom, 6 School of Medicine, University of Liverpool, Liverpool, United Kingdom, 7 Primary Care, Population Science and Medical Education, Faculty of Medicine, University of Southampton, Southampton, United Kingdom, 8 Nuffield Department of Primary Care Health Sciences, University of Oxford, Oxford, United Kingdom, 9 Department of Primary Care and Public Health, School of Public Health, Imperial College London, London, United Kingdom, 10 Faculty of Life Sciences and Medicine, King's College London, London, United Kingdom

* edward.meinert@newcastle.ac.uk

**Data Availability Statement:** Deidentified individual participant data has been deposited in the Open Science Framework (OSF) repository

## Abstract

Childhood obesity is a growing global health concern. Although mobile health apps have the potential to deliver behavioural interventions, their impact is commonly limited by a lack of sufficient engagement. The purpose of this study was to explore barriers and facilitators to engagement with a family-focused app and its perceived impact on motivation, self-efficacy, and behaviour. Parents with at least one child under 18 and healthcare professionals working with children were recruited; all participants were allocated to use the NoObesity app over a 6-month period. The mixed-methods design was based on the Non-adoption, Abandonment, Scale-Up, Spread, and Sustainability and Reach, Effectiveness, Adoption, Implementation, and Maintenance frameworks. Qualitative and quantitative data were gathered through semi-structured interviews, questionnaires, and app use data (logins and in-app self-reported data). 35 parents were included in the final analysis; quantitative results were analysed descriptively and thematic analysis was conducted on the qualitative data. Key barriers to engagement were boredom, forgetting, and usability issues and key barriers to potential impact on behaviours were accessibility, lack of motivation, and family characteristics. Novelty, gamification features, reminders, goal setting, progress monitoring and feedback, and suggestions for healthy foods and activities were key facilitators to engagement with the app and behaviours. A key observation was that intervention strategies could help address many motivation and capability barriers, but there was a gap in strategies addressing opportunity barriers. Without incorporating strategies that successfully mitigate barriers in all three determinants of behaviour, an intervention is unlikely to be successful. We

(DOI: 10.17605/OSF.IO/RUQ4G). The dataset does not include the full demographic data collected to avoid indirect identification of participants. Following the guidance in Hrynaszkiewicz et al. (2010), we have included only 3 indirect identifiers: age, gender, and ethnicity. Two participants did not consent for their data to be made available for future studies, so that data has been removed from the available dataset.

**Funding:** MMI and EM received research grant funding for this study from the former Health Education England, which is now the South East School of Public Health, Workforce Training and Education Directorate, NHS England (grant reference number: AM1000393). EM and MMI are supported by the NIHR Newcastle Biomedical Research Council (BRC). The views expressed in the paper belong to the authors and not necessarily those of the South East School of Public Health, NHS England, the University of Plymouth, the University of Liverpool, the University of Southampton, the University of Oxford, Newcastle University, the NIHR Newcastle BRC, or Imperial College London. The funding body had no editorial control and was not involved in the decision to submit the article for publication.

**Competing interests:** ER and AP are employees of the former Health Education England (now the South East School of Public Health, Workforce Training and Education Directorate, NHS England) and were involved in the development of the NoObesity apps. ER and AP contributed qualitative data and reviewed the final manuscript prior to submission, but the academic authors retained editorial control. MMI and EM received research grant funding for this study from the former Health Education England, which is now the South East School of Public Health, Workforce Training and Education Directorate, NHS England (grant reference number: AM1000393). EM and MMI are supported by the NIHR Newcastle Biomedical Research Council (BRC). The views expressed in the paper belong to the authors and not necessarily those of the South East School of Public Health, NHS England, the University of Plymouth, the University of Liverpool, the University of Southampton, the University of Oxford, Newcastle University, the NIHR Newcastle BRC, or Imperial College London. The funding body had no editorial control and was not involved in the decision to submit the article for publication. The other authors have declared that no competing interests exist.

highlight key recommendations for developers to consider when designing the features and implementation of digital health interventions.

**Trial Registration:** ClinicalTrials.gov (NCT05261555).

## Author summary

Childhood obesity is a public health concern worldwide, but healthcare services lack the capacity to provide support and advice for all families on strategies for how it can be prevented or mitigated. This study explores what factors influence families' engagement with a mobile health app for childhood obesity prevention and their perceptions of its impact on their physical activity and eating behaviours. We found that novelty and interactivity–making the app fun and interesting–were key features that helped motivate families to keep using the app. Features of the app that enabled families set goals, keep track of their progress, and get suggestions for activities and healthy eating helped support their motivation and belief in their ability to engage in healthier behaviours. Beyond motivation, many families faced barriers related to their opportunity to engage in healthy behaviours such as lack of time, safe outdoor spaces, or affordable healthy food options. These findings highlighted the importance of understanding what prevents people from engaging with digital health interventions and the target behaviour change so that we can design interventions that help mitigate those barriers.

## Introduction

Childhood obesity is a global concern affecting around 1 in 5 children worldwide [1]. Being overweight can negatively impact physical and mental health [2–8] and strain healthcare resources [9]. Determinants of childhood obesity are complex, including structural inequalities and genetics [10–12], but individuals can influence behavioural contributors like diet and activity [7,13–15]. The scope of the problem of childhood obesity in the UK—over a third of 10–11 year old children are overweight or obese [16]—requires support beyond the delivery capacity of clinical services. Mobile apps are a promising tool to support behaviour change because of their ubiquity [17–20], but their impact is commonly limited by low engagement [21–26].

Previous studies have supported the potential benefit of mobile apps for weight management for adults and children [18,27–33], especially for highly engaged users [27], but many interventions lack robust evidence of a long-term impact on weight [19,31,34–36]. It is essential that mobile health apps achieve sufficient user engagement to deliver impact [27,28,37]. Strategies for engaging users with digital interventions for childhood obesity are being examined [38], but further understanding of how to mitigate barriers to engagement is needed [39–42]. Behaviour Change Techniques (BCTs), the "smallest active ingredients" of interventions that can act to influence behaviour [43], are increasingly being incorporated into mobile health apps. Understanding how individual and contextual factors influence theoretical strategies, like BCTs, that aim to support engagement and behaviour change is necessary to optimise their behavioural and health impacts [44]. The purpose of this study was to gain insight into the context behind how and why parents engaged and disengaged with an app for childhood obesity, how it influenced their families' behaviour, and how potential improvements could be incorporated. Specifically, the study aimed to 1) generate barriers and facilitators that

influence engagement with the app and 2) to evaluate the app's impact on parents in terms of perceived motivation, self-efficacy, weight-related health behaviours, and communication with healthcare professionals (HCPs) [45].

## Methods

### Study design

We conducted a mixed-methods interventional study based on a Phase 1 implementation science design (Fig 1). Phase 1 studies focus on generating evidence around the implementation strategies and other factors that could influence acceptability, effectiveness, and successful adoption [46,47]. This is a critical step in the evaluative process, as it informs intervention refinement to improve impact in later efficacy- and effectiveness-oriented evaluations. The primary analysis examined qualitative data about barriers and facilitators to users' engagement with the app and its impact on motivation, self-efficacy, and behaviours. To improve the credibility of the qualitative results, they were triangulated with quantitative survey and app use data [45,48].

Three theoretical frameworks informed the study: the Reach Effectiveness Adoption Implementation Maintenance (RE-AIM) framework [49], the Non-adoption, Abandonment, Scale-up, Spread and Sustainability (NASSS) framework [50], and the Capability, Opportunity, Motivation—Behaviour (COM-B) model [51,52]. RE-AIM and NASSS informed the data collection plan, to capture a comprehensive set of individual- and system-level factors (Table 1 [49,50,53,54]). The COM-B model was used to analyse how various factors influenced engagement and behaviour [55–59]. The SRQR [60] (S1 Table) and TREND [61] checklists (S2 Table) were used to ensure completeness of reporting.

### Intervention

The NoObesity app was developed by Health Education England (HEE) to prevent and manage childhood obesity. The term HEE is used as that was the organisation at the time of the study, but the organisation has since been incorporated into the South East School of Public Health, Workforce Training and Education Directorate, NHS England. The target audience included families wanting behavioural support (regardless of weight) and HCPs working with families [62]. It was not based on a particular behavioural theory but was co-produced with HCPs [62] and included features aligned with several BCTs (see S3 Table for a full list of

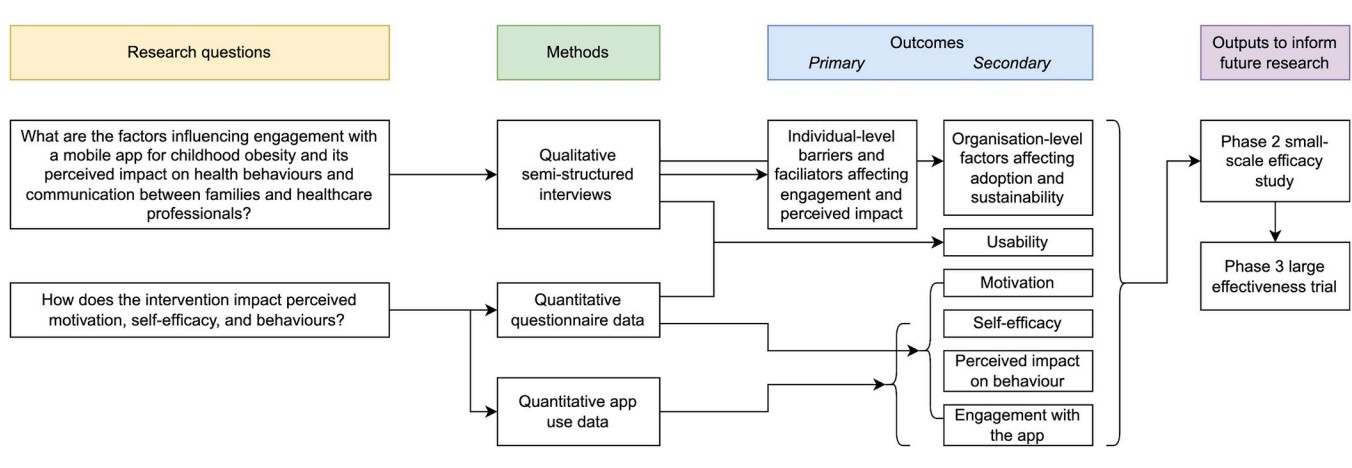

**Fig 1. Study logic diagram.**

**Table 1. Alignment of RE-AIM and NASSS frameworks (based on [49,50,53,54]).**

| NASSS domains | RE-AIM dimensions | Key considerations | Key outcomes |
|---|---|---|---|
| **Domain 1: The condition or illness** Characteristics of users (clinical, socio-cultural, etc.) | *N/A* | • Domain 1 emphasises the importance of understanding contextual factors and how they influence implementation | • Sample characteristics (demographics) • Barriers and facilitators |
| **Domain 2: The technology** Features of technology associated with usability, acceptability, trustworthiness, and sustainability | *Implementation* Fidelity of intervention delivery (system), use of intervention strategies (user) | • Both focus on key issues of usability, acceptability, and other factors associated with individual use • RE-AIM captures issue of whether the intervention was delivered as intended | • System Usability Scale • Key themes about usability, acceptability, and app features • Barriers and facilitators • Suggestions for improvement |
| **Domain 3: The value proposition** Evidence of impact and value, for users and suppliers (incl. safety, efficacy, benefit, affordability) | *Effectiveness* Impact on intended outcomes | • Both emphasise the importance of evidence to support adoption of intervention (safety, positive impact on intended outcomes) • NASSS considers impact of intervention for suppliers as well as users, and how that factors into the intervention's sustainability | • Key themes about motivation and perceived impact on self-efficacy and behaviours • Ratings of app motivation strategies • Self-reported success at goals (in app and survey) • Self-reported self-efficacy (in app and survey) |
| **Domain 4: The adopter system** Adoption (and continued use) by staff and users; reasons for non-adoption / abandonment; assumptions about use (capability, opportunity) | *Adoption* Number, proportion, and representativeness of places/people willing to adopt intervention (system) *Reach* Number, proportion, and representativeness participants (user) | • RE-AIM 'adoption' also relates to the potential added value • NASSS includes adoption by patients, staff, and setting, which is captured separately in RE-AIM ('Reach' and 'Adoption') • Key questions around adoption: how much, how long, why, why not, what is required? | • Participation and drop-out data • Key themes about adoption of HCP link and impact of name on adoption • Ratings of HCP link and app name |
| **Domain 5: The organization** Organization's capacity and readiness for adoption and scale-up; recognition of need for and availability of dedicated resources; disruption to routines; work to implement | *N/A* | • Domains 5 and 6 both related to various elements of the adoption and maintenance of the intervention, so have been analysed independently from the RE-AIM framework | • Key themes from qualitative interview with HEE representatives |
| **Domain 6: The wider context** Political, economic, regulatory, professional, sociocultural factors | *N/A* | | |
| **Domain 7: Embedding and adaptation over time** Feasibility of ongoing adaptation; organisational resilience (willingness to reflect and adapt) | *Maintenance* Institutionalisation of intervention in routine processes (system), long-term impact on outcomes (user) | • Both capture institutional / organisational maintenance • RE-AIM also captures individual maintenance • In this study, the organisational aspects of ongoing maintenance were closely aligned with key themes in Domains 5 and 6, and were grouped with them in the analysis | • Self-reported intention to continue use of app • Self-reported length of use (if discontinued) • App use data from system (length of use, most recent use date) |

features, previously published in [45]). For families, the app helps them set behavioural goals to improve healthy eating and physical activity (e.g. cut down on snacks, walk to school, get the kids outside), track progress on their goals, and learn about healthy behaviours together through games and suggestions. For HCPs, the app provides training and tools for how to communicate effectively with families about childhood obesity (Fig 2). At the time of the study, the app was publicly available in the Apple App Store and Google Play store and promoted on HEE's website.

## Sample and recruitment

From March to August 2020, a convenience sample of parents and HCPs living anywhere in the UK was recruited using social media advertising (Google Adwords, Instagram Ads) [63]. Participants self-selected to use the app in their daily lives over a 6-month intervention period

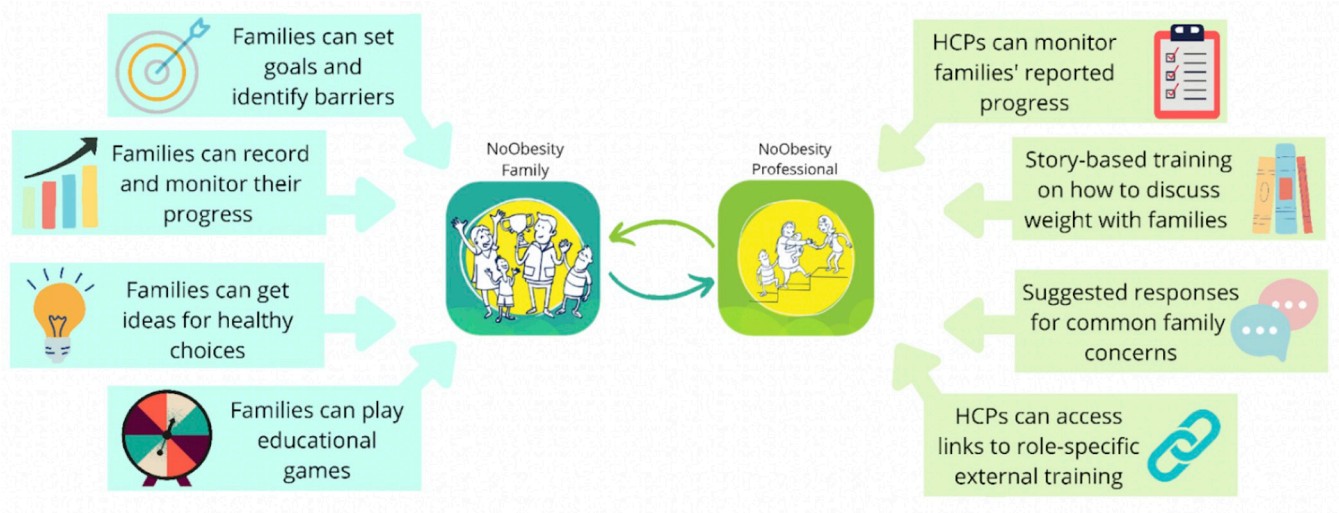

**Fig 2. NoObesity Family and Professional app features.**

based on the following inclusion criteria: (1) a parent/guardian (with at least one child any age under 18 years) OR a HCP working with families, (2) willing and able to use the app, (3) fluent in English, and (4) no previous use of the app. Individuals who were unable or unwilling to provide informed consent, had hearing impairments, or were known to the research team were excluded. Eligibility was not limited by weight to reflect real-world app use and the broad target audience. Informed consent was collected from all participants on enrolment using a Qualtrics survey. We also recruited two employees from HEE (ER, AP) who helped develop NoObesity to capture organisational-level factors influencing adoption and sustainability (Table 1).

Stratified random sampling (based on gender, ethnicity, and income) was used to invite a subset of 20 participants for semi-structured interviews (SSIs) from the 61 participants who completed the demographic questionnaire. Interviewed participants and those included in the quantitative data analysis (completed all questionnaires and passed an attention check question) received a £100 (USD$139) Amazon voucher.

## Procedure

Ethical approval was obtained from the University of Oxford Medical Sciences Interdivisional Research Ethics Committee (R62092/RE001) and the University of Plymouth's Faculty Research Ethics and Integrity Committee (19/20-1316). Thirty-minute SSIs were conducted on Microsoft Teams (S1 Text) and transcribed by a third-party company (Rev). To mitigate response bias, participants were informed that the interviewer was independent from HEE and had not been involved in intervention development. Three questionnaires were administered online using Qualtrics: consent, demographics based on OxIS 2019 [64] (September 2020), and the final questionnaire at the end of the 6-month intervention period (April 2021), which examined usability, use, and impact on motivation, self-efficacy, and behaviour and included attention check questions (S1 Text). App use data was collected in June 2021.

## Outcomes

The primary outcome—barriers and facilitators to engagement and health behaviours—was explored through the qualitative SSIs. Secondary outcomes (Table 2) were examined through

**Table 2. Quantitative secondary outcome measures.**

| Outcome | Outcome measure |
|---|---|
| Engagement with the app | App-captured dates of first and last logins |
| | Quantitative self-reported app use duration |
| Engagement with the study | Uptake (number of participants recruited) |
| | Dropout rates (number of participants completing each stage of the data collection) |
| Usability | System Usability Scale (SUS, scale: 0–100) [65]; completed by parents twice—once for themselves and once from their children's perspective |
| Motivation | Quantitative ratings of the impact of specific features on motivation (5-point Likert scale, see Table 3 for details) |
| Self-efficacy | Bandura's self-efficacy scale [66] (confidence in ability to do goal activities; scale: 0–100) |
| | In-app self-reported confidence in ability to achieve goals (6 star rating scale) |
| Perceived impact on behaviour | Quantitative questionnaire items about physical activity and healthy eating behaviours (5-point Likert scale, see Table 3 for details) |
| | In-app self-reported goal progress data |
| Opinions about app features | Quantitative questionnaire items about app features (5-point Likert scale, see Table 3 for details) |

qualitative interviews, questionnaire, and app use data (including self-reported data and objective login data captured via the app admin portal).

## Analysis

Qualitative data was analysed using a codebook thematic analysis approach [67,68] in Dedoose (version 9.0.17) [69]. To improve credibility—the alignment of participants' feedback with our representation of their feedback—four authors engaged with the data (first independently, then collaboratively) over a prolonged period. This ensured that individual preconceptions were challenged and alternative interpretations considered [70]. An initial list of codes was generated inductively by one author based on the transcripts, which were coded at the semantic level (interpreting only the explicit meaning of the participants' words) [71]. This codebook was provided to three independent authors who deductively coded the data (generating additional codes if needed). Codings then evolved through intensive collaboration between the authors. Themes and sub-themes were generated by one author as topic summaries [68] and then discussed and refined by all coders (S4 Table).

From the initial thematic framework, factors influencing engagement and perceived motivation, self-efficacy, and behaviours were deductively mapped to the COM-B model (S5 Table) [51,52,72]. A codebook for the mapping was created using the COM-B components [52] and the Theoretical Domains Framework (TDF) [73], which captures more detailed subthemes of the COM-B components (Table 3) and examples from the literature to guide coding in a digital health context (S6 Table) [58,59]. These factors are the main outcomes discussed. Quantitative survey and app use data were analysed using descriptive statistics. Where possible, these outcomes were triangulated with qualitative data.

## Results

### Sample characteristics

Consented participants included 225 parents/guardians and 6 HCPs. 61 parents completed the demographic survey; 15 of whom were interviewed. Almost 40% of parents (85/225) completed the final questionnaire, but only 16% (35/225) met the criteria to be included in the

**Table 3. COM-B and TDF components and definitions [52,73].**

| COM-B | Definition [52] | TDF | Definition [73] |
|---|---|---|---|
| Physical capability | Physical skill, strength or stamina | Skills | An ability or proficiency acquired through practice |
| Psycho- logical capability | Knowledge or psychological skills, strength or stamina to engage in the necessary mental processes | Knowledge | An awareness of the existence of something |
| | | Memory, attention, and decision processes | The ability to retain information, focus selectively on aspects of the environment and choose between two or more alternatives |
| | | Behavioural regulation | Anything aimed at managing or changing objectively observed or measured actions |
| Physical opportunity | Opportunity afforded by the environment involving time, resources, locations, cues, physical 'affordance' | Environ- mental context and resources | Any circumstance of a person's situation or environment that discourages or encourages the development of skills and abilities, independence, social competence and adaptive behaviour |
| Social opportunity | Opportunity afforded by interpersonal influences, social cues and cultural norms that influence the way that we think about things, e.g. the words and concepts that make up our language | Social influences | Those interpersonal processes that can cause individuals to change their thoughts, feelings, or behaviours |
| Reflective motivation | Reflective processes involving plans (self- conscious intentions) and evaluations (beliefs about what is good and bad) | Social / professional role and identity | A coherent set of behaviours and displayed personal qualities of an individual in a social or work setting |
| | | Beliefs about capabilities | Acceptance of the truth, reality or validity about an ability, talent or facility that a person can put to constructive use |
| | | Optimism | The confidence that things will happen for the best or that desired goals will be attained |
| | | Beliefs about consequences | Acceptance of the truth, reality, or validity about outcomes of a behaviour in a given situation |
| | | Intentions | A conscious decision to perform a behaviour or a resolve to act in a certain way |
| | | Goals | Mental representations of outcomes or end states that an individual wants to achieve |
| Automatic motivation | Automatic processes involving emotional reactions, desires (wants and needs), impulses, inhibitions, drive states and reflex responses | Reinforce- ment | Increasing the probability of a response by arranging a dependent relationship, or contingency, between the response and a given stimulus |
| | | Emotion | A complex reaction pattern, involving experiential, behavioural, and physiological elements, by which the individual attempts to deal with a personally significant matter or event |

quantitative analysis: completing demographic and final surveys and passing an attention check. No HCPs completed the study (Fig 3). Most participants were female and white, with a mean age of 39 years (SD 6.0; Table 4). Of the participants invited for interview via stratified random sampling, 8 responded and 5 completed the interviews. After a second round of stratified random sampling, all eligible participants were invited to reach a sample of 15 interviewed participants. Participants who did not complete follow-up assessments may still have been using the app, as it was publicly available and usage was not controlled.

## Engagement with the intervention (adoption and use)

A large sample was initially recruited (n = 225), suggesting that there was interest in the app, but there was substantial drop-out from the study and discontinued app use (captured via self-report and app login data). Although most interviewed participants reported no plans to stop using NoObesity, only half of final survey respondents (19/35, 54%) reported still using the app and of the participants from whom app use data was collected (n = 26), only a third (9/26, 34%) had recently logged in by or after the end of the study (April 2021). On average, there were 5.1 months between account setup and most recent login (95% CI: 3.9–6.3, range: 0.0–

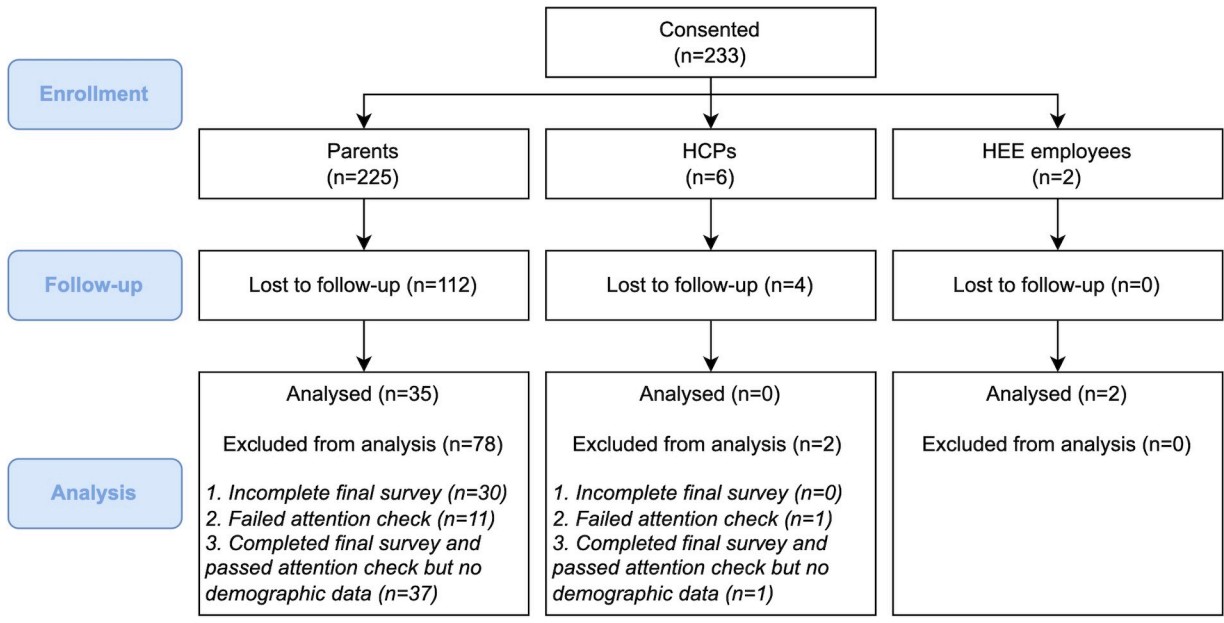

**Fig 3. Participant flow diagram (HCPs: healthcare professionals; HEE: Health Education England).**

10.4 months). This section examines the key capability, opportunity, and motivational factors identified from the thematic analysis as influencing engagement with the intervention (S5 Table). These could be grouped into two themes: motivation and usability (Figs 4–5).

### Factors influencing motivation to engage

Most of the factors that influenced participants' motivation to engage related to the automatic and reflective motivation components of COM-B, although several overlapped with psychological capability (eg. TDF domain: 'memory, attention, decision processes, and knowledge') and physical opportunity (eg. TDF domain: 'environmental context' offered by the app). Novelty and variety were key for engagement and demonstrate this overlap; without something new to explore, participants felt bored, had reduced intentions to open the app, and needed increased cognitive effort to overcome these barriers (Box 1). Participants all valued feedback, but several found the app's feedback lacking or unhelpful. Visualisations of progress were suggested to help support motivation and more interactivity and gamification (e.g. daily

---

**Box 1. Sample of participant quotes about factors affecting engagement with the app**

*"Once you've gone through the sections it has, then there's nothing else you can really do with it. So there's not much interaction . . . to keep you engaged."* (ppt 2)

*"Another reason why I didn't really do it is because. . . it feels quite repetitive having to click . . . Tuesday, Wednesday, Thursday, Friday [to record family progress] for every single challenge."* (ppt 3)

*"Had there been some new content on there, it would have encouraged me to come back more and more and go, "Oh, what more can I learn?""* (ppt 7)

---

**Table 4. Participant characteristics of the sample who completed the final quantitative survey (n = 35) and interviewed sample (n = 15).**

| Characteristic | Number (%) *final survey* | Number (%) *interviewed* |
|---|---|---|
| Gender | | |
| Female | 31 (89) | 14 (93) |
| Male | 4 (11) | 1 (7) |
| Age (parent/guardian) | | |
| 30–39 | 20 (57) | 11 (73) |
| 40–49 | 12 (34) | 3 (20) |
| 50–59 | 3 (9) | 1 (7) |
| Ethnicity | | |
| White | 32 (91) | 12 (80) |
| Asian | 2 (6) | 1 (7) |
| Black | 1 (3) | 2 (13)* |
| Highest completed qualification | | |
| Bachelor's degree or higher | 23 (66) | 8 (53) |
| Lower than Bachelor's degree | 12 (34) | 7 (47) |
| Annual household income before tax | | |
| £12,500–19,999 | 3 (9) | 1 (7) |
| £20,000–29,999 | 5 (14) | 2 (13) |
| £30,000–39,999 | 7 (20) | 5 (33) |
| £40,000–49,999 | 5 (14) | 2 (13) |
| £50,000–59,999 | 4 (11) | 2 (13) |
| £60,000–69,999 | 2 (6) | 0 (0) |
| £70,000–79,999 | 1 (3) | 0 (0) |
| ≥ £80,000 | 6 (17) | 1 (7) |
| Prefer not to say | 2 (6) | 2 (13) |
| Employment | | |
| Working full time (30 hours a week or more) | 12 (34) | 3 (20) |
| Working part time (8–29 hours a week) | 15 (43) | 7 (47) |
| Doing housework, looking after children or other persons | 4 (11) | 3 (20) |
| Student | 2 (6) | 1 (7) |
| Permanently sick or disabled | 2 (6) | 1 (7) |
| Location | | |
| A country village | 5 (14) | 2 (13) |
| A small city or town | 24 (69) | 10 (67) |
| The suburbs or outskirts of a big city | 4 (11) | 1 (7) |
| A big city | 2 (6) | 2 (13) |
| Number of adults in household | | |
| One | 3 (9) | 2 (13) |
| Two | 31 (89) | 13 (87) |
| Three or more | 1 (3) | 0 (0) |
| Number of kids in household | | |
| One | 11 (31) | 8 (53) |
| Two | 16 (46) | 5 (33) |
| Three | 6 (17) | 2 (13) |
| Four | 2 (6) | 0 (0) |

*Not all interviewed participants completed the final quantitative survey

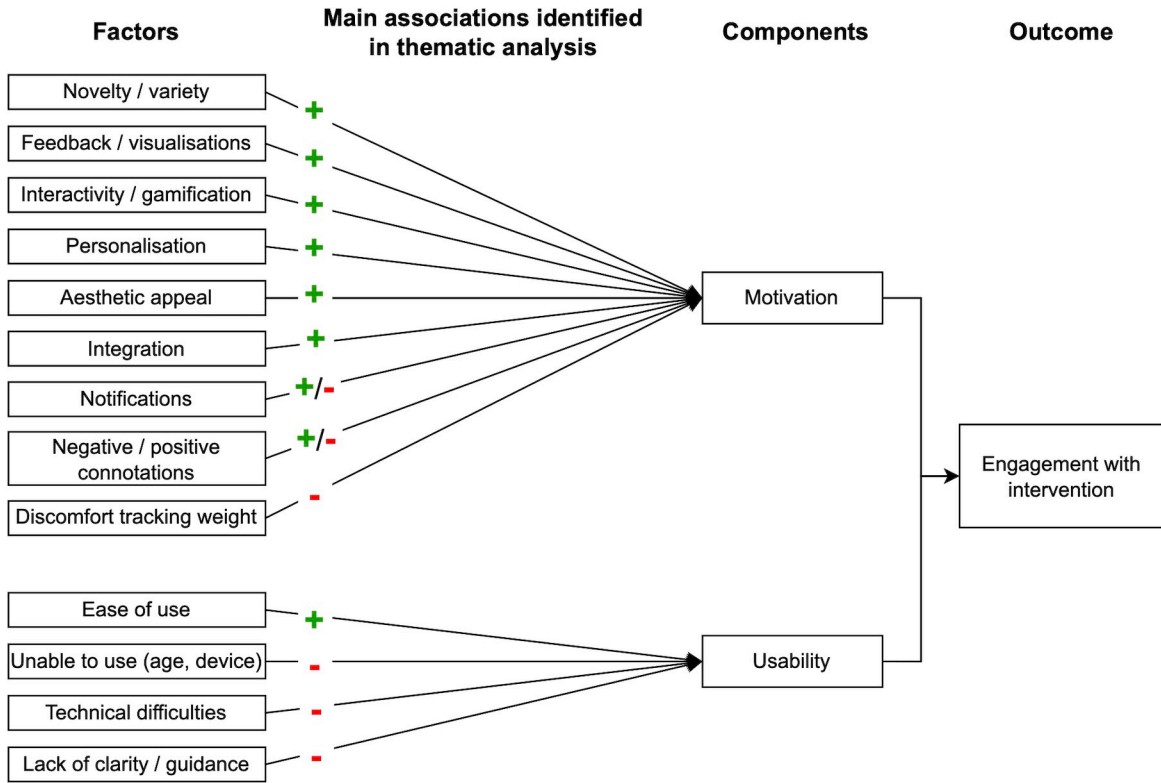

**Fig 4. Summary of factors identified in the thematic analysis and how they were associated with engagement (Note: + and—signs refer to whether the factor had a positive or negative influence on the outcome).**

challenges, tangible rewards, and friendly competition) to help engage children. Opinions about notifications were mixed: many participants appreciated the reminder, but several found them annoying or not delivered at appropriate times. A couple participants did not like the app's aesthetic and felt this hindered use.

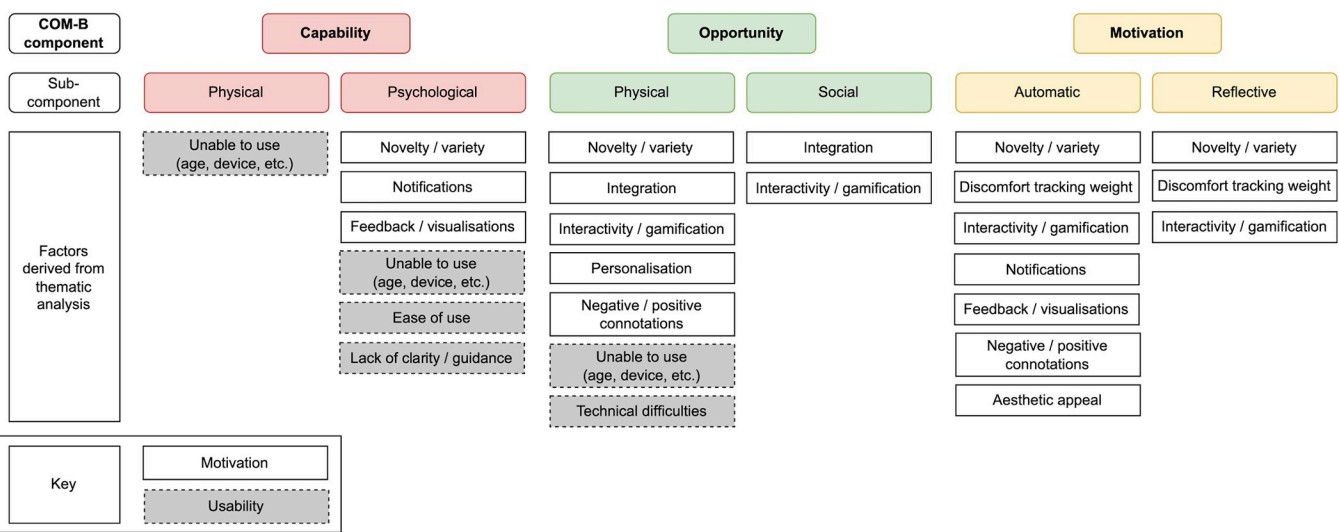

**Fig 5. Summary of key factors for engagement mapped to the COM-B components.**

Positivity and a focus on behaviour rather than weight were important; some parents highlighted that while they might track calories or weight for themselves, they felt uncomfortable doing so for their children. This was also reflected in opinions of the name "NoObesity;" although most participants did not personally feel strongly about it, they thought it had negative connotations and that a more positive focus on health behaviours could improve new user engagement. The survey results supported this; participants reported somewhat disliking the name (M = 2.6/5, 95% CI: 2.2–3.0).

## Factors associated with usability

Factors associated with ability to engage were primarily related to psychological capability and physical opportunity. Overall, interviewees found the app "quite easy to use," in line with SUS scores (M = 70.3/100, 95% CI: 63.8–76.8), although ratings were slightly lower when parents completed the SUS from their child(ren)'s perspective (M = 65.1/100, 95% CI: 58.3–71.9). Barriers were primarily related to the technology (app freezing) and a lack of clarity and guidance in the app. Several parents reported not understanding the target audience, how to report goal progress as a family unit if only some of the family members had successfully achieved the goal, or how the app's reward system worked, which hindered its potential impact on motivation (Box 2).

> ## Box 2. Participant quotes about factors affecting usability
>
> *"Because of the iconography and how it looks, I think it's quite easy to use. It's quite self-explanatory."* (ppt 4)
>
> *"It could've possibly done with an introduction page to start with."* (ppt 6)
>
> *"There's a lot of external links. I think you could quite easily get lost looking and looking for things."* (ppt 13)
>
> *"Two of us have gone out for a walk, but two of us haven't. Do I score it [as having completed the family goal], do I not?"* (ppt 11)

## Perceived impact on behaviours

Perceived impact on health behaviours was mixed and often conflicting; most participants reported some progress on their goals but many said that the app did not have an impact on their behaviour (Box 3). In-app self-reported goal progress indicated that families were successful at achieving any particular goal for a mean of 7.5 weeks (95% CI: 5.5–9.5), a median of 3 weeks, and a mode of 0 weeks (n = 27). Factors relating to all three COM-B components were associated with the perceived impact on families' motivation, self-efficacy, and health behaviours (Figs 6–7).

## Facilitators for behaviours

Factors relating to perceived impact on motivation and self-efficacy were mostly related to motivation and psychological capability. Many participants found goal setting and progress recording motivating because it increased their awareness of their current behaviour and provided a sense of accountability (Box 4). Having "*achievable, attainable*" goals, "*being able to*

Box 3. Participant quotes about perceived impact of the app

*"The healthier meals . . . last week we were able to tick off three or four, but I know we did more than that, it was just the actual ones I ticked off."* (ppt 15)

*"[I] don't necessarily think there were any changes. Let's say, we live very close to school, so it's easiest to walk. So that thing that I put in as our first goal about walking to school every day. We were going to use it every day, anyway."* (ppt 4)

*input data,*" and getting ideas for goal setting, physical activity, and healthy eating supported self-efficacy; which was largely aligned with the quantitative ratings of impact of app features on motivation, which were highest for 'doing something together as a family', 'suggestions for activities and healthy eating', and 'goal setting' (Table 5). Other facilitators included prompts and notifications and useful feedback (from app or a linked HCP). Median self-efficacy scores on the survey were around 70% (70/100) (range: 20–100%), similar to the mean confidence (out of 6 stars) users self-reported when setting goals on the app (3.9/6 stars, ~65%).

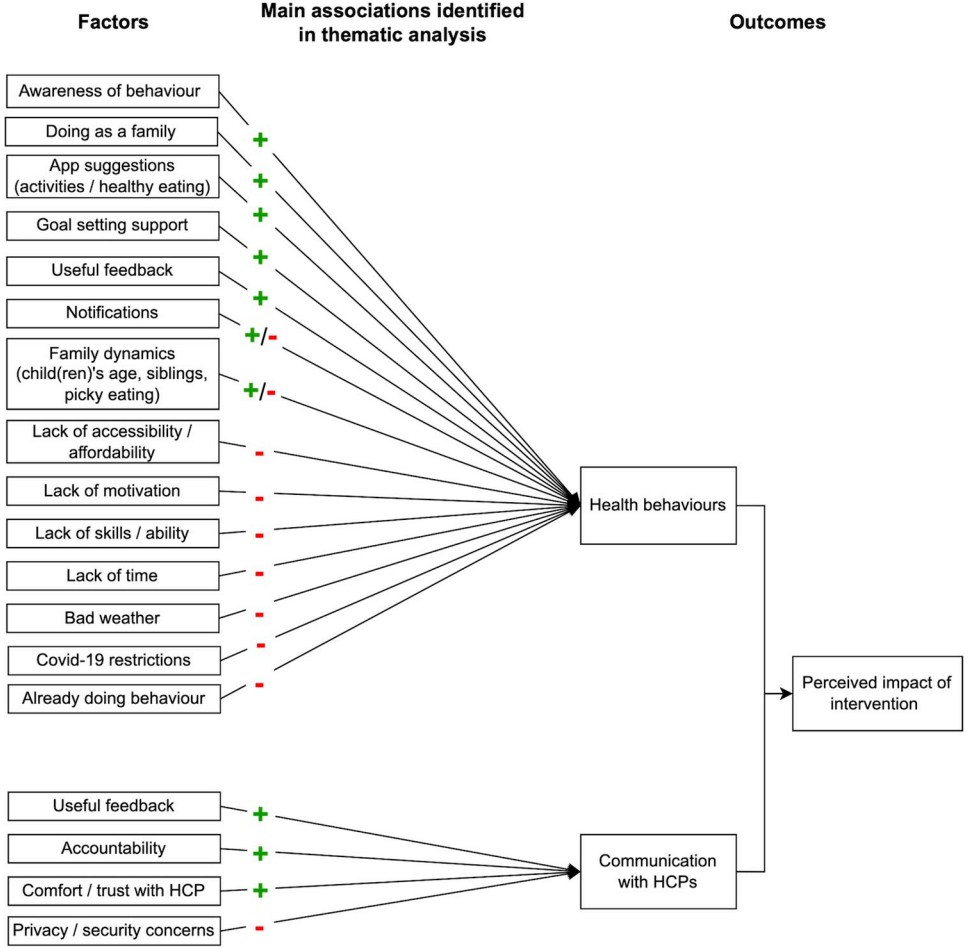

**Fig 6. Summary of factors identified in the thematic analysis associated with the app's perceived impact.**

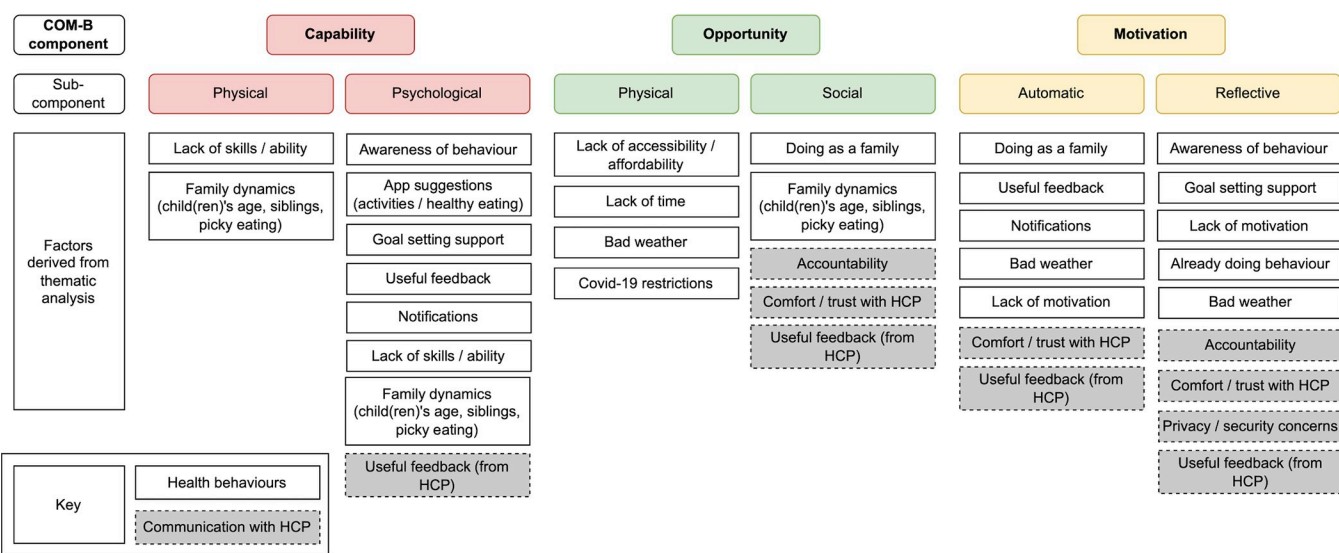

**Fig 7. Summary of key factors for perceived impact mapped to the COM-B components.**

### Barriers to behaviours

Most behavioural barriers were associated with physical opportunity, including Covid-19 restrictions, time, weather, accessibility, and affordability (Box 5). There were also barriers relating to social opportunity (children's willingness to try new foods and sibling teasing) and physical and psychological capability (barriers relating to children's age and skills). Some participants suggested potential capability barriers, such as not knowing how to cook a healthy meal, but few experienced these barriers themselves. A lack of motivation was also mentioned; sometimes related to physical opportunity barriers (e.g. outdoor activity in bad weather), but not always with a reason given.

> ## Box 4. Participant quotes about factors affecting motivation and self-efficacy
>
> *"This kind of brings it forward for you to actually acknowledge what you're doing and what you're not doing, and to help you to correct it. So it was like, I'm not going to have a snack today because I want to say that I managed to not have a snack today."* (ppt 2)
>
> *"I would have liked to be able to see trends. . . it's nice to have a look at your graph, isn't it?"* (ppt 1)
>
> *"If you've downloaded it and it's a family challenge, and it's telling you to do it all together, it does make it easier, I think"* (ppt 3)
>
> *"I thought the useful links were good . . . because if you're not an outdoorsy family, you might not even know where to start"* (ppt 8)
>
> *"Without [feedback from a HCP], it was completely lacking. [The app] didn't give me feedback on whether my goals were good ideas. It didn't give me feedback on how well I was doing meeting my goals."* (ppt 1)

**Table 5. Participant ratings of the app and its features' perceived impact (n = 35).**

| Item | Mean (/5) | St. dev. | 95% CI | Likert scale points meaning (1–5) |
|---|---|---|---|---|
| **App's support for behaviour** | | | | |
| Set goals | 3·5 | 1·4 | 3·0–4·0 | Strongly disagree—strongly agree |
| Achieve goals | 3·5 | 1·0 | 3·1–3·8 | Strongly disagree—strongly agree |
| Eat healthier | 3·4 | 1·3 | 3·0–3·8 | Strongly disagree—strongly agree |
| Be more active | 3·3 | 1·3 | 2·9–3·8 | Strongly disagree—strongly agree |
| **Perceived impact of app on motivation** | | | | |
| Eat healthier | 2·7 | 1·1 | 2·3–3·0 | Not at all—extremely effective |
| Be more active | 2·6 | 1·1 | 2·2–3·0 | Not at all—extremely effective |
| **Perceived impact of motivation strategy** | | | | |
| Doing something together as a family | 3·2 | 1·3 | 2·8–3·6 | Not at all—extremely effective |
| Suggestions of activities | 2·9 | 1·2 | 2·5–3·3 | Not at all—extremely effective |
| Suggestions for healthier eating | 2·8 | 1·0 | 2·5–3·2 | Not at all—extremely effective |
| Goal-setting | 2·8 | 1·3 | 2·4–3·3 | Not at all—extremely effective |
| Self-monitoring | 2·7 | 1·1 | 2·4–3·1 | Not at all—extremely effective |
| Points / trophies[a] | 2·7 | 1·3 | 2·2–3·1 | Not at all—extremely effective |
| Notifications[a] | 2·3 | 1·3 | 1·9–2·8 | Not at all—extremely effective |
| Feedback about current behaviour | 2·2 | 0·9 | 1·9–2·5 | Not at all—extremely effective |
| Games | 2·2 | 1·2 | 1·8–2·6 | Not at all—extremely effective |
| Family photo | 1·9 | 1·3 | 1·5–2·3 | Not at all—extremely effective |
| Linking to a HCP | 1·6 | 0·9 | 1·3–1·9 | Not at all—extremely effective |
| **Opinion of features** | | | | |
| Points / trophies | 3·5 | 1·1 | 3·1–3·8 | Dislike a great deal—like a great deal |
| Self-monitoring | 2·7 | 1·0 | 2·4–3·1 | Not at all—extremely useful |
| Information feedback | 2·6 | 1·1 | 2·3–3·0 | Not at all—extremely useful |
| **Opinion on healthcare provider link** | | | | |
| Level of comfort | 3·1 | 1·4 | 2·6–3·6 | Extremely uncomfortable—extremely comfortable |
| Perceived usefulness | 2·8 | 1·4 | 2·4–3·3 | Not at all—extremely useful |
| Likelihood of linking | 2·5 | 1·4 | 2·0–2·9 | Extremely unlikely—extremely likely |

[a]Sample was missing one data point (n = 34)

---

## Box 5. Participant quotes about factors affecting behaviours

*"We don't have outside space for us. So, it's an effort to do it every day."* (ppt 1)

*"Some of the ideas were good, but there's just nowhere around here that we'd be able to do them."* (ppt 10)

*"Our time is very limited. I go out to work and I'm not back till half five/six. So once we've had a family meal it's dark, the children aren't wanting to go back out. I'm not really wanting to go back out. It's cold."* (ppt 15)

*"It's just easier to buy junk food, it's cheaper and easier to put. . . something in the oven rather than cook from scratch. And some people might not necessarily have the skills."* (ppt 8)

## Perceived impact on communication

No HCPs completed the study and no participants connected with a HCP via the app (most had not seen a HCP during the intervention period because of Covid-19). Hypothetically, most parents felt comfortable communicating via the app and that the HCP link could provide useful feedback and accountability (Box 6), although a couple raised concerns about the app not reflecting true behaviour, bothering the HCP, discomfort discussing a sensitive topic, or data security.

---

### Box 6. Participant quotes about factors affecting communication with HCPs

*"I would have appreciated a real person saying, 'well, that's a good goal, or that's a bit wishy-washy. Perhaps you could make that better'."* (ppt 1)

*"It would be good because I think for me personally, if I'm accountable for something then, somebody checking in on you and going to look at what progress you've made. . .it would make me more determined to do it"* (ppt 8)

*"I don't really feel I've got a relationship with my GP. So for me, I think I felt a bit reluctant about doing that just because I don't know them."* (ppt 11)

---

## Organisational factors

Interviews with HEE employees identified organisational-level factors that could influence the app's sustainability. The app's focus fits within UK priority areas around childhood obesity and digital-first healthcare. Its holistic approach, combining workforce development and service delivery, spans the mandate of several governmental bodies (since the interviews, HEE itself has been incorporated into NHS England) [74]. The HEE employees felt that this could be a benefit, as inter-agency collaboration could provide access to more expertise, but that it creates potential ambiguity around ownership, which could be a risk for longer-term maintenance.

## Discussion

### Main findings

This study used the COM-B model to explore factors associated with engagement and the perceived impact of a digital health app for childhood obesity and the NASSS and RE-AIM frameworks to generate insights for its implementation. Facilitators of engagement included novelty and variety, gamification and feedback, and a clear and positive tone. In terms of health behaviours, app features including BCTs [43] such as goal setting, problem solving (suggestions for goals), self-monitoring, feedback on behaviour, and instruction on how to perform the behaviour (suggestions for healthy foods and activities) helped support motivation and capability. Key barriers—including accessibility (e.g. Covid-19 restrictions, weather, affordability, and availability) and family characteristics—were largely related to opportunity, highlighting a gap in the app's engagement and behaviour change strategies.

### Recommendations in the context of existing evidence

There are many digital health interventions that have been developed and evaluated to target childhood obesity, but while studies often assess user perceptions and acceptability, there is

limited investigation of factors influencing engagement with mobile health apps for childhood obesity [31,75]. A recent evaluation of the Aim2Be app for childhood obesity identified different patterns of engagement and associated them with demographic characteristics, but not user perceptions [76], and another conducted a formative co-design process with parents to identify preferences for engagement and barriers to the health behaviours, which were largely similar to those identified here; however, these were not analysed in depth in that paper [77]. For this reason, we examine our findings in comparison to factors influencing engagement with digital health interventions more generally.

All three COM-B components are important for digital health intervention engagement and impact [78–85]. Facilitators for engagement—a simple visually-appealing interface, feedback and visualisations of progress, guidance, customisation, reminders, access to a HCP, and positive messaging—were aligned with previous recommendations in the literature [59,86–89]. Several factors, including common engagement strategies like gamification [90–92], notifications [82,93], and competition [94], could be barriers or facilitators. For example, in-app points and trophies alone may not be sufficiently meaningful rewards to support motivation [92] and notifications' impact can depend on their timing, frequency, and personalisation [82,93,95,96]. Friendly competition could help engage children by increasing fun and motivation [59]; however, parents were concerned about teasing and negative self-image. Social comparison can be demotivating [94], but this could be mitigated through team-based competition that de-emphasises the individual [42,97]. For childhood obesity, enabling intergroup family-team-based competition could help support engagement and focus on positive behaviours rather than weight.

Key facilitators for behavioural impact included awareness of behaviour, accountability, motivation, prompts, suggestions, family characteristics, and accessibility. Features that support a family focus [97], goal-setting, progress recording, positive messaging, feedback, and suggestions and inspiration could help support perceived motivation and self-efficacy in this context, in line with meta-analytic evidence in adults [98]. The relationship between motivation and accountability has been previously observed [99,100] and theoretically linked to adherence [101]. Accountability has the potential to help support behaviours not done for enjoyment [101,102], so more active involvement of a linked HCP with families via the app may have helped increase engagement with the intervention and behavioural goals; however, this may not work for everyone, as some participants worried that their app-reported progress did not reflect real behaviour and did not want to feel judged.

Key structural- and individual-level barriers were also largely aligned with existing evidence [81,85,89,103,104]. Structural inequities can hinder weight management and contribute to feelings of stigma or blame that reduce motivation [105,106]; mitigating these barriers (e.g. by highlighting affordable healthy meal options or locating free resources personalised to the users' location) will be necessary to enable positive health behaviours. Individual-level barriers were more aligned with motivation than capability [51], perhaps because families less aware of weight issues may have been less likely to use an app called "NoObesity." As a key facilitator for engagement was the app's family focus, we recommend that developers consider the psychology behind their users' motivation to engage and how they can frame their intervention as something enjoyable rather than necessary but onerous [107]. Overall, we recommend that digital intervention designers research specific barriers (especially opportunity barriers) for their target populations and behaviours to identify the most appropriate means of mitigating them in that context. Table 6 highlights additional recommendations for intervention development, through the lens of the NASSS and RE-AIM implementation frameworks.

**Table 6. Summary of implications for implementing similar interventions in the future.**

| NASSS domains | RE-AIM dimensions | Insights for designing interventions for successful implementation |
|---|---|---|
| Domain 1: The condition or illness | N/A | • Social determinants of health—such as income, environment, employment, etc.—are important factors that affect families' abilities to engage in healthy weight-related behaviours.<br>• Digital health interventions should account for these contextual differences in users by incorporating advice and behavioural suggestions that meet their varying abilities and financial and environmental circumstances.<br>• For example, this could include providing advice or recipes on how to eat healthy on a tight budget and including suggestions of physical activities that can be engaged in within a house if outdoor spaces are unavailable or inaccessible. |
| Domain 2: The technology | Implementation | • The importance of novelty as a factor to support engagement raises a potential issue for not-for-profit organisations that do not have capacity or funding for regular updates and uploading of new content for an intervention.<br>• These types of limitations should be accounted for in the intervention design, for example, by setting up the intervention to provide content periodically over the course of the intervention or by enabling users to generate content (where appropriate).<br>• Specific implementation strategies should be developed in collaboration with the target population, to ensure that the intervention meets their needs and circumstances. |
| Domain 3: The value proposition | Effectiveness | • To have a strong case to support public health interventions, evidence of positive impact is key—on an individual level, users should be able to see evidence of impact for themselves (eg. through progress monitoring and feedback from the intervention, or prompted self-reflection); on an organisational level, there needs to be significant evidence of positive impact to support widespread adoption.<br>• Evaluations of effectiveness should include assessments of engagement as a prerequisite for impact and should include end-user perspectives on what key outcomes they consider desirable (e.g. weight, mood, perceived energy, ability to do certain activities). |
| Domain 4: The adopter system | Adoption | • Given the high demand on HCPs' time and changes in practices and routines associated with implementation, perceived value, familiarity, and ease of use will be essential to clinical adoption [83].<br>• User-centred design processes should include a variety of clinical stakeholders, including those who would be directly engaging with the intervention and those responsible for managing adoption and implementation processes [108].<br>• This will be needed to identify and address challenges to adoption, whether they are related to perceived value and desirability of the intervention or structural barriers such as its lack of integration into existing routines |
| | Reach | • Social media and app store advertisement is unlikely to be sufficient to reach the target population in many cases; to increase reach, organisations should diversify dissemination by engaging with community groups and settings, specific to their target population's clinical and demographic characteristics.<br>• The use of user-centred design with the target population could also improve reach by ensuring that, as well as being aware of the intervention, the target population expects that it will add value. |
| Domain 5: The organisation<br>Domain 6: The wider context | N/A | • Within governmental and healthcare organisations, political changes could provide a potential challenge for continuity and responsibility of interventions; solutions for such issues will be case dependent, but should be considered from the outset to ensure there is a plan for sustainability and maintenance. |
| Domain 7: Embedding and adaptation over time | Maintenance | • Digital health interventions that aim to change behaviour contributing to health outcomes should use technical and social features that facilitate the incorporation of these behaviours into regular routines (see recommendations for features above). |

## Strengths, limitations, and future research

Strengths of this study included the use of mixed methods, theoretical frameworks (which guided the investigation of a comprehensive set of factors influencing adoption and implementation), and independent and collaborative coding by several authors for a rigorous thematic analysis. Due to time and resource limitations, mapping to the COM-B model was only completed by one author, which limits the robustness of that aspect of the analysis.

The main limitation was the sample; high dropout and non-representative demographics [109,110] created potential bias. Stratified random sampling was used to improve the diversity of the SSIs, but the limited overall sample and the need to invite all eligible participants to complete 15 interviews meant that interviewees' perspectives are likely to represent a particular set of lived experiences, reducing generalisability. The convenience method of sampling via social media also introduces potential bias, as participants who respond are likely to be those with

the strongest opinions about the intervention and the reach of the recruitment materials may not reflect a representative sample of the population. Random sampling could reduce this potential bias, although it may not reflect real-world engagement. Alternate methods of recruitment may have more success in increasing the sample size; for example, recruiting families and HCPs via schools, children's centres, primary care centres, and other community organisations. A larger-scale study with a more representative sample will be needed to evaluate the impact of the intervention on health behaviour change and childhood obesity. We also recommend that future studies capture demographic data immediately after consent, to enable analysis of participant characteristics in the initial and final samples.

Other limitations included the exclusion of children (to facilitate data collection and ethical approval) and the monetary reward for participants who completed the study (which might have incentivised app engagement). Data about the age of the children was not collected in the demographic data, but this was highlighted as an important factor in the qualitative analysis for consideration in future design and evaluation. Additional factors that could influence engagement and perceived impact are initial behavioural- and weight-related characteristics; these were not assessed as weight was not an inclusion criteria but may be an important factor to consider in future research.

The lack of data from HCPs meant we could not evaluate their perceptions of the intervention or their reasons for not participating in the study. The intervention period took place during the Covid-19 pandemic; unusually high demands on HCP's time might have precluded engagement with the study and intervention, but lack of participation could also be related to a lack of awareness of the study or HCP perceptions of the app. This will be an important area for future investigation, as previous research in digital health has suggested that integrating human support can improve engagement and is important for childhood obesity management [111,112].

Our ability to triangulate qualitative and quantitative data was also limited; the app use data that the system could record was relatively minimal and did not enable detailed examination of the frequency, intensity, time, or type of engagement (first and last login only, rather than more detailed data capturing number of logins or use of intervention features) [48]. The app is being redesigned based on theoretical frameworks and the findings of this study; this process will also improve its ability to capture app usage data. The limitations described here will be addressed in future larger-scale efficacy and effectiveness studies.

## Conclusions

The growing prevalence of childhood obesity and the overwhelming demand for healthcare resources more generally has resulted in a need for easily accessible interventions to empower families to manage weight-related health. The benefits of digital health interventions are often limited by insufficient engagement, so understanding facilitators and barriers to engagement is essential. This paper highlights how the use of a theoretical behavioural framework can clarify key barriers to engagement with a digital health intervention and its target behaviour(s) and suggest mitigations. We recommend that digital intervention designers incorporate interactivity, novel content and suggestions, goal setting and progress monitoring, feedback and accountability, reminders, guidance on how to use the app, personalisation, and a positive and visually-appealing design. The caveat is that developers must identify whether these suggested intervention strategies align with engagement barriers and patient or other factors in their particular context; if strategies address capability and motivation barriers but not opportunity barriers, the intervention is unlikely to succeed. In terms of implementation, the NASSS and RE-AIM domains highlight key, interconnected factors that can influence the success of an

intervention—particularly important is the need to develop and demonstrate a value proposition that meets the needs and circumstances of the target users and clinical adopting systems, which can best be executed by adopting user-centred design practices to ensure that solutions to potential barriers are incorporated from early stages in development.

## Supporting information

**S1 Text. Semi-structured interview guide and study questionnaires.**
(DOCX)

**S1 Table. SRQR checklist.**
(DOCX)

**S2 Table. TREND checklist.**
(DOC)

**S3 Table. NoObesity Family and Professional apps functional overview (adapted from protocol) [45].**
(DOCX)

**S4 Table. Thematic framework.**
(DOCX)

**S5 Table. Thematic framework mapping factors to COM-B.**
(DOCX)

**S6 Table. COM-B and TDF coding framework with digital health examples.**
(DOCX)

## Author Contributions

**Conceptualization:** Em Rahman, Alison Potter, Wendy Lawrence, Michelle Helena van Velthoven, Edward Meinert.

**Formal analysis:** Madison Milne-Ives, Hannah Bradwell, Rebecca Baines, Timothy Boey.

**Funding acquisition:** Em Rahman.

**Investigation:** Madison Milne-Ives.

**Methodology:** Em Rahman, Alison Potter, Wendy Lawrence, Michelle Helena van Velthoven, Edward Meinert.

**Supervision:** Edward Meinert.

**Visualization:** Madison Milne-Ives.

**Writing – original draft:** Madison Milne-Ives.

**Writing – review & editing:** Madison Milne-Ives, Em Rahman, Hannah Bradwell, Rebecca Baines, Timothy Boey, Alison Potter, Wendy Lawrence, Michelle Helena van Velthoven, Edward Meinert.

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
