## [Decision Letter · Decision Letter 0]

5 Dec 2023

PDIG-D-23-00411

Barriers and facilitators to engagement with and impact of a childhood obesity app: a mixed-methods study

PLOS Digital Health

Dear Dr. Meinert,

Thank you for submitting your manuscript to PLOS Digital Health. After careful consideration, we feel that it has merit but does not fully meet PLOS Digital Health's publication criteria as it currently stands. Therefore, we invite you to submit a revised version of the manuscript that addresses the points raised during the review process.

Please submit your revised manuscript within 60 days Feb 03 2024 11:59PM. If you will need more time than this to complete your revisions, please reply to this message or contact the journal office at digitalhealth@plos.org. Please include the following items when submitting your revised manuscript:

We look forward to receiving your revised manuscript.

Kind regards,

Haleh Ayatollahi

Section Editor

PLOS Digital Health

Journal Requirements:

Additional Editor Comments (if provided):

Reviewers' comments:

Reviewer's Responses to Questions

**Comments to the Author**

1. Does this manuscript meet PLOS Digital Health’s publication criteria? Is the manuscript technically sound, and do the data support the conclusions? The manuscript must describe methodologically and ethically rigorous research with conclusions that are appropriately drawn based on the data presented.

Reviewer #1: Partly

Reviewer #2: Yes

Reviewer #3: Yes

Reviewer #4: Yes

Reviewer #5: Yes

2. Has the statistical analysis been performed appropriately and rigorously?

Reviewer #1: N/A

Reviewer #2: No

Reviewer #3: N/A

Reviewer #4: N/A

Reviewer #5: Yes

3. Have the authors made all data underlying the findings in their manuscript fully available (please refer to the Data Availability Statement at the start of the manuscript PDF file)?

Reviewer #1: No

Reviewer #2: Yes

Reviewer #3: Yes

Reviewer #4: Yes

Reviewer #5: No

4. Is the manuscript presented in an intelligible fashion and written in standard English?

Reviewer #1: Yes

Reviewer #2: Yes

Reviewer #3: Yes

Reviewer #4: Yes

Reviewer #5: Yes

5. Review Comments to the Author

Reviewer #1: Please follow data sharing and availability rules for PLOS Digital Health. It is not permissible to put the corresponding author as the primary point of contact for data sharing----there are ways around institutional restrictions.

Reviewer #2: Regarding the checklist, we aim to provide clarity to the authors and acknowledge their efforts in advance.

TITLE and ABSTRACT

Clarification is needed on how units were allocated to interventions.

Under the section "Information on target population or study sample":

The article lacks information on the target population. As it deals with a diverse group, specifically the pediatric population, details about age need to be provided as it represents a distinct patient group. Additionally, the article doesn't cover location or ethnicity. It's suggested to include information regarding the age range of children and parents since the study measures facilitators. Considering the variability in age among participants, it's crucial to account for this variable.

A structured abstract is recommended:

It includes a structured summary that meets the description.

Scientific background and rationale

The provided explanation appears rational for the objective of the work presented. The primary objectives of the work are also clearly stated.

Eligibility criteria for participants

The primary concern observed pertains to the external validity of the work. The sampling method used is convenience sampling rather than stratified. The target population is widely dispersed. As the ultimate goal is to generalize results in the pediatric population, this study includes only 35 subjects without describing the characteristics of the intervened pediatric population. Furthermore, with a wide age range (30 years) within the subject group, randomization could better generalize most of these factors. However, in this study, they are included through diffusion and participation, leading to significant selection bias. Inclusion of voluntary participants implies self-selection bias, limiting generalization to only those willing to participate, thus not reflecting the broader population.

Method of recruitment

Once again, the text doesn't mention a randomized method for registration and participant selection. Self-selection bias is noteworthy, where respondents with specific characteristics are more inclined to participate, potentially skewing the results. Respondents' inclinations towards certain topics lead to unintended biases affecting the study's rationality. Voluntary response bias further limits representation to only those with strong opinions, disregarding others. Ethical constraints prevented an assessment of this bias. Selecting parents from a nearby community through random sampling would have been advisable to achieve a more representative sample.

Details of interventions

The article specifies a series of questionnaires, which are appended and considered relevant. The authors adequately address these aspects.

Ethical considerations

The article shows records of the ethics committee R62092/RE001.

Data collection instruments and technologies

A comprehensive description is provided, considered appropriate.

Specific objectives and hypotheses

Clear specific objectives are outlined. However, the achievability of the third objective seems questionable given the study's design. Omitting this objective is suggested due to the speculative nature of both the results and the study's context and methodology.

Summary of results and statistical design

For each primary and secondary outcome, a summary of results for each estimation study condition, along with the estimated effect size and confidence intervals indicating precision, were adequately presented. The statistical design for the proposed model is considered appropriate.

Integration with prior work and implications

In the discussion, it would be beneficial to compare these results with previous studies, citing another intervention in a similar context and its identified domains and impacts.

Final Considerations:

Acknowledging this as a commendable piece of work, restructuring known weaknesses is necessary. While redoing the work may not be feasible, revising objectives and methods to better align with the proposed objectives is crucial. Mere theorization cannot suffice as a response to an objective. Reviewing the text content with the suggested adjustments and addressing the comments upon resubmission is recommended.

Reviewer #3: It is an interesting manuscript analyzing barriers and facilitators to engagement with an app designed to impact on children obesity.

The authors report that only 16% of the parents that gave their consentient participated in the study mainly because they didn’t meet the inclusion criteria. However, they do not suggest any strategy that could be applicable in future studies to increase the sample. The authors designed the study including the participation of three important groups, parents, HCP and HEE. One of the limitations of the study is that no HCP participated. The authors do not discus neither clearly indicate the weight of this limitation in the lack of engagement (only 34% of the sample used the app at the end of the study). The HCPs could have played an important role in the parents' commitment to use the app and so this could have been a very important factor in the results. 

Even though the authors included appropriate references they do not give evidence of the originality of the study, because they do not compare their approach or results with other studies

Reviewer #4: This well-written manuscript presents an evaluation of a weight management intervention for children that is used in the family context. It uses data collected via a mixed methods approach to study barriers and facilitators to intervention use and behaviour change. Grounding the evaluation in multiple theories and frameworks and providing helpful visualisations of the qualitative findings are clear strengths of the manuscript. There are a few open questions that might need to be addressed prior to publication:

1) Why was engagement with app assessed via first and last login and not via the number of logins or use of intervention features?

2) To further understanding inclusivity of the intervention it might be insightful - if the data is available - to compare the final sample to the enrolled sample in terms of demographics.

3) The recommendations provided in Table 6 are helpful, but not necessarily rooted in the presented data (e.g., many recommendations focus on HCPs which could not be studied). Please include references to the literature for all recommendations that were derived from the literature instead of from the data at hand.

4) It's a shame that HCPs did not take part in the evaluation, and that they were generally not contacted by the app users. What are the reasons for HCPs not being more involved in the evaluation, and how could exchange between HCPs and users be promoted? This might be especially crucial since there is first evidence that including personal contact in apps might promote engagement and effectiveness compared to digital-only services, and that in person interventions are generally more effective than digital weight loss interventions:

Dombrowski, S., Whitcomb, E., Olthuis, J., Witherspoon, R., & Hebert, J. (2021). Syetemactire review of RCTs comparing face-to-face vs remote behaviour change interventions for weight management. In 35th Conference of the European Health Psychology Society.

Mamalaki, E., Poulimeneas, D., Tsiampalis, T., Kouvari, M., Karipidou, M., Bathrellou, E., ... & Yannakoulia, M. (2022). The effectiveness of technology‐based interventions for weight loss maintenance: a systematic review of randomized controlled trials with meta‐analysis. Obesity Reviews, 23(9), e13483.

Reviewer #5: The paper aimed to explore barriers and facilitators to engagement with a family-focused app for childhood obesity prevention and its perceived impact on motivation, self-efficacy, and behaviour. The topic is relevant and up-to-date, and the findings highlighted the importance of examining how a digital health intervention could be designed and implemented. Although the method session, especially the sampling and recruitment needs some clarification, the study is robust, well-written, and provides important insights for future interventions and scaling-up. Some minor comments are suggested below:

Title: Since the paper assessed the perceived impact on motivation, self-efficacy, and behaviour, not the intervention impact on childhood obesity per se, it is suggested to include “perceived impact” instead of only “impact” in the title for clarification. How it is described may lead the reader to think the paper evaluated the impact of the app intervention on childhood obesity. 

On page 6, session “Intervention”, the authors stated that: “The target audience included families wanting behavioural support (regardless of weight)”. Please, explain a little more about how the app is routinely used. How are the users targeted, and how can they reach support with the app? Is the app publicly available? Is there any publicity on the app so that families who want behavioural support know how to seek it? 

On page 7, session “Sample and recruitment”, what was the justification for the exclusion of “children and vulnerable adults”? What was defined as a vulnerable population? 

On page 8 “Stratified random sampling (based on gender, ethnicity, and income) was used to select a subset of parents for semi-structured interviews (SSIs)”. Was it out of the 61 parents who completed the demographic survey? How was the final number of 15 interview parents reached? What were the proportions of each sampling strata?

On page 12, session “Sample characteristics”. It is important to clarify in the protocol people who were lost during the follow-up, and those, who were excluded due to incomplete data (but still finish the study). If participants could complete the study regardless of if they filled out the initial forms, then, they were not lost for follow-up. Please, review the flow diagram. 

On page 12, please clarify if a participant would still use the app after consent, but not complete any of the questionnaires. How was this controlled? Is there any information about the initial sample of parents (225)? The sample is a bit confusing. Please, clarify it. It would be interesting to see some initial characteristics for the total population (225) to see if they differ from people who were not lost for follow-up.

Page 14 – “only half of final survey respondents (19/35, 54%) reported still using the app and only a third had recently logged in by the end of the study (9/26, 34%)”. It is not clear what this “a third” means (9/26). Where do these 26 people come from, if there were 35 interviewees, and 26 is not a third of 35? 

Page 14 – “On average, there were 5.1 months between account setup and most recent login (95% CI: 3.9-6.3, range: 0.0-10.4 months).” How is the maximum time 10.4 months if the intervention duration was 6 months, and only individuals with no previous use of the app were recruited?

Page 17: “Perceived impact on health behaviours was mixed and often conflicting; most participants reported some progress on their goals, but many said that the app did not have an impact on their behaviour.” The paper needs to make it clear how is the app designed to interact with parents on the aim of behaviour changes to overcome childhood obesity in the family context. The way it is presented, it is not clear how these goals and changes reported by parents are aligned with the main app goal. It would be helpful to describe some of the goals, and how they are applied in the family context, not the individual. 

Page 27: Limitations. Please, address the limitations regarding the exclusion of vulnerable people, the method of recruitment (social media advertising), and how this could introduce bias in the sample.

6. PLOS authors have the option to publish the peer review history of their article (what does this mean?). If published, this will include your full peer review and any attached files.

**Do you want your identity to be public for this peer review?** For information about this choice, including consent withdrawal, please see our Privacy Policy.

Reviewer #1: No

Reviewer #2: Yes: Adan Pacifuentes-Orozco

Reviewer #3: No

Reviewer #4: No

Reviewer #5: No

---

## [Decision Letter · Decision Letter 1]

28 Dec 2023

PDIG-D-23-00411R1

Barriers and facilitators to engagement with and perceived impact of a childhood obesity app: a mixed-methods study

PLOS Digital Health

Dear Dr. Meinert,

Thank you for submitting your manuscript to PLOS Digital Health. After careful consideration, we feel that it has merit but does not fully meet PLOS Digital Health's publication criteria as it currently stands. Therefore, we invite you to submit a revised version of the manuscript that addresses the points raised during the review process.

Please submit your revised manuscript within 60 days Feb 26 2024 11:59PM. If you will need more time than this to complete your revisions, please reply to this message or contact the journal office at digitalhealth@plos.org. Please include the following items when submitting your revised manuscript:

We look forward to receiving your revised manuscript.

Kind regards,

Haleh Ayatollahi

Section Editor

PLOS Digital Health

Journal Requirements:

Additional Editor Comments (if provided):

Reviewers' comments:

Reviewer's Responses to Questions

**Comments to the Author**

1. If the authors have adequately addressed your comments raised in a previous round of review and you feel that this manuscript is now acceptable for publication, you may indicate that here to bypass the “Comments to the Author” section, enter your conflict of interest statement in the “Confidential to Editor” section, and submit your "Accept" recommendation.

Reviewer #2: All comments have been addressed

Reviewer #3: All comments have been addressed

Reviewer #4: All comments have been addressed

Reviewer #5: All comments have been addressed

2. Does this manuscript meet PLOS Digital Health’s publication criteria? Is the manuscript technically sound, and do the data support the conclusions? The manuscript must describe methodologically and ethically rigorous research with conclusions that are appropriately drawn based on the data presented.

Reviewer #2: No

Reviewer #3: Yes

Reviewer #4: No

Reviewer #5: Yes

3. Has the statistical analysis been performed appropriately and rigorously?

Reviewer #2: Yes

Reviewer #3: N/A

Reviewer #4: Yes

Reviewer #5: N/A

4. Have the authors made all data underlying the findings in their manuscript fully available (please refer to the Data Availability Statement at the start of the manuscript PDF file)?

Reviewer #2: Yes

Reviewer #3: Yes

Reviewer #4: Yes

Reviewer #5: Yes

5. Is the manuscript presented in an intelligible fashion and written in standard English?

Reviewer #2: (No Response)

Reviewer #3: Yes

Reviewer #4: Yes

Reviewer #5: Yes

6. Review Comments to the Author

Reviewer #2: The main concern observed persists, which relates to the external validity of the work. The sampling method used is convenience sampling rather than stratified. The target population is widely dispersed. Since the ultimate goal is to generalize the results to the pediatric population, this study includes a small number of subjects without describing the characteristics of the intervened pediatric population, which is an important factor. Upon author review, it is commented once again that the corrections were only made in a theoretical manner. I believe that due to the type of sampling, we cannot generalize the results as stated in the primary objective. It is suggested to increase the sample size or modify the title to be more specific to the scope of the external validity of the work. No corrections were made regarding this point, so I suggest once again modifying the scope of the work or specifying objectives that may be more achievable with the proposed design.

Reviewer #3: It's great to hear that the authors addressed the questions and comments effectively, incorporating changes into the manuscript. Although the sample size and the absence of healthcare professionals (HCPs) in the study are acknowledged as limitations, the reviewer recognizes the study's importance in providing valuable insights. The focus on factors like ethnicity, income, education, and familiar environment for understanding the impact of apps on weight loss in children with obesity is highlighted as a key contribution. The reviewer believes that the substantial changes made by the authors meet the standards for publication in Plos Digital Health

Reviewer #4: (No Response)

Reviewer #5: The authors fully addressed the reviewers' comments, clarifying the questions and/or changing the manuscript. Most comments were included in the new version, which enhanced the overall quality of the manuscript and is now in alignment with the points raised by the reviewers. Considering the relevance and design of the study, the potential to contribute to the field, and the overall quality of the paper after this revision, I am recommending acceptance.

7. PLOS authors have the option to publish the peer review history of their article (what does this mean?). If published, this will include your full peer review and any attached files.

**Do you want your identity to be public for this peer review?** For information about this choice, including consent withdrawal, please see our Privacy Policy. 

Reviewer #2: Yes: 

Reviewer #3: Yes: Cleva Villanueva

Reviewer #4: No

Reviewer #5: No

---

## [Decision Letter · Decision Letter 2]

22 Jan 2024

PDIG-D-23-00411R2

Barriers and facilitators to engagement with and perceived impact of a childhood obesity app: a mixed-methods study

PLOS Digital Health

Dear Dr. Meinert,

Thank you for submitting your manuscript to PLOS Digital Health. After careful consideration, we feel that it has merit but does not fully meet PLOS Digital Health's publication criteria as it currently stands. Therefore, we invite you to submit a revised version of the manuscript that addresses the points raised during the review process.

Please submit your revised manuscript within 60 days Mar 22 2024 11:59PM. If you will need more time than this to complete your revisions, please reply to this message or contact the journal office at digitalhealth@plos.org. Please include the following items when submitting your revised manuscript:

We look forward to receiving your revised manuscript.

Kind regards,

Haleh Ayatollahi

Section Editor

PLOS Digital Health

Journal Requirements:

Additional Editor Comments (if provided):

Reviewers' comments:

Reviewer's Responses to Questions

**Comments to the Author**

1. If the authors have adequately addressed your comments raised in a previous round of review and you feel that this manuscript is now acceptable for publication, you may indicate that here to bypass the “Comments to the Author” section, enter your conflict of interest statement in the “Confidential to Editor” section, and submit your "Accept" recommendation.

Reviewer #2: All comments have been addressed

2. Does this manuscript meet PLOS Digital Health’s publication criteria? Is the manuscript technically sound, and do the data support the conclusions? The manuscript must describe methodologically and ethically rigorous research with conclusions that are appropriately drawn based on the data presented.

Reviewer #2: Yes

3. Has the statistical analysis been performed appropriately and rigorously?

Reviewer #2: Yes

4. Have the authors made all data underlying the findings in their manuscript fully available (please refer to the Data Availability Statement at the start of the manuscript PDF file)?

Reviewer #2: Yes

5. Is the manuscript presented in an intelligible fashion and written in standard English?

Reviewer #2: Yes

6. Review Comments to the Author

Reviewer #2: Dear author, we find ourselves in the same situation once again. The comments regarding statistics and models are clear and understood, and I agree with the methodology. However, I believe that clarity in the title is of utmost importance in a study, as one of the fundamental principles of research is simply the alignment of the title with the objectives.

The responses I am requesting are simply related to the modification of the title based on the actual objectives you are presenting. To illustrate my point, let me provide an example:

The current title reads: Title: Barriers and facilitators to engagement with and perceived impact of a childhood obesity app: a mixed-methods study However, the objectives you present are different, as follows:

The purpose of this study was to gain insight into why why people engaged and disengaged with an app for childhood obesity, how it influenced their behaviour, and how this could be improved. Specifically, it aimed to 1) generate barriers and facilitators that influence users’ engagement with the app and 2) to evaluate its impact on perceived motivation, self-efficacy, weight-related health behaviours, and communication between healthcare professionals (HCPs) and families

However, in your responses, you use these objectives: "To describe participants' experiences, key factors, and commonalities influencing engagement and perceived impact of a digital health app." This does not align with your general title, as the title suggests that we need to examine how people, including children, engage with the app. Following this logic, it assumes that children are not considered individuals in this analysis. This raises an ethical issue that needs to be addressed.

I suggest changing your objectives as follows:

1. Analyze how some parents engage and disengage with a childhood obesity app, and how it influences their behavior and potential improvements in app usage.

2. Evaluate the impact on parents in terms of perceived motivation, self-efficacy, weight-related behaviors, and communication with healthcare professionals.

Additionally, you mention: "For example, in our study, we were more interested in the generalization (or transferability) of the factors identified as associated with participation and perceived impact, as this has the potential to shape the design of digital interventions in the specific context of childhood obesity and other health contexts."

However, we cannot discuss a health context without considering the subjects of the study, as these factors always influence the perception of the phenomenon. For example, I cannot assess the quality of life perception in cancer patients without knowing the cancer stage, the patient's age, and the histological type of cancer, as these factors are crucial in interventions.

In conclusion, the requested changes were not made, and the responses remain theoretical. The main issue lies in the title and the objectives, which are not clearly aligned with what is presented in the responses. In future instances, I urge you to be more specific in your objectives, as they determine the methodology to be followed. The title and objectives must be specific to avoid confusion. I appreciate your work and understanding.

7. PLOS authors have the option to publish the peer review history of their article (what does this mean?). If published, this will include your full peer review and any attached files.

**Do you want your identity to be public for this peer review?** For information about this choice, including consent withdrawal, please see our Privacy Policy. 

Reviewer #2: No

---

## [Decision Letter · Decision Letter 3]

6 Feb 2024

PDIG-D-23-00411R3

Barriers and facilitators to parents' engagement with and perceived impact of a childhood obesity app: a mixed-methods study

PLOS Digital Health

Dear Dr. Meinert,

Thank you for submitting your manuscript to PLOS Digital Health. After careful consideration, we feel that it has merit but does not fully meet PLOS Digital Health's publication criteria as it currently stands. Therefore, we invite you to submit a revised version of the manuscript that addresses the points raised during the review process.

Please submit your revised manuscript within 30 days Mar 07 2024 11:59PM. If you will need more time than this to complete your revisions, please reply to this message or contact the journal office at digitalhealth@plos.org. Please include the following items when submitting your revised manuscript:

We look forward to receiving your revised manuscript.

Kind regards,

Haleh Ayatollahi

Section Editor

PLOS Digital Health

Journal Requirements:

Additional Editor Comments (if provided):

Reviewers' comments:

Reviewer's Responses to Questions

**Comments to the Author**

1. If the authors have adequately addressed your comments raised in a previous round of review and you feel that this manuscript is now acceptable for publication, you may indicate that here to bypass the “Comments to the Author” section, enter your conflict of interest statement in the “Confidential to Editor” section, and submit your "Accept" recommendation.

Reviewer #2: All comments have been addressed

2. Does this manuscript meet PLOS Digital Health’s publication criteria? Is the manuscript technically sound, and do the data support the conclusions? The manuscript must describe methodologically and ethically rigorous research with conclusions that are appropriately drawn based on the data presented.

Reviewer #2: Yes

3. Has the statistical analysis been performed appropriately and rigorously?

Reviewer #2: Yes

4. Have the authors made all data underlying the findings in their manuscript fully available (please refer to the Data Availability Statement at the start of the manuscript PDF file)?

Reviewer #2: Yes

5. Is the manuscript presented in an intelligible fashion and written in standard English?

Reviewer #2: Yes

6. Review Comments to the Author

Reviewer #2: I am pleased to see that you have made modifications to the text. I believe that delineating the work allows for a better understanding of it. I consider it important to address a couple of necessary issues prior to publication, for which I suggest communicating with the participants:

Describe the general characteristics of the children under parental care.

Provide information about the parents' level of education.

Share the average usage time of the application.

All of the above is necessary to provide us with a better context of the situation. While I do believe the article has potential, I recommend making these changes before considering it suitable for publication.

7. PLOS authors have the option to publish the peer review history of their article (what does this mean?). If published, this will include your full peer review and any attached files.

**Do you want your identity to be public for this peer review?** For information about this choice, including consent withdrawal, please see our Privacy Policy. 

Reviewer #2: Yes: Adán Pacifuentes-Orozco

---

## [Decision Letter · Decision Letter 4]

28 Feb 2024

Barriers and facilitators to parents' engagement with and perceived impact of a childhood obesity app: a mixed-methods study

PDIG-D-23-00411R4

Dear Dr Meinert,

We are pleased to inform you that your manuscript 'Barriers and facilitators to parents' engagement with and perceived impact of a childhood obesity app: a mixed-methods study' has been provisionally accepted for publication in PLOS Digital Health.

Best regards,

Haleh Ayatollahi

Section Editor

PLOS Digital Health

Reviewer Comments (if any, and for reference):

Reviewer's Responses to Questions

**Comments to the Author**

1. If the authors have adequately addressed your comments raised in a previous round of review and you feel that this manuscript is now acceptable for publication, you may indicate that here to bypass the “Comments to the Author” section, enter your conflict of interest statement in the “Confidential to Editor” section, and submit your "Accept" recommendation.

Reviewer #2: All comments have been addressed

2. Does this manuscript meet PLOS Digital Health’s publication criteria? Is the manuscript technically sound, and do the data support the conclusions? The manuscript must describe methodologically and ethically rigorous research with conclusions that are appropriately drawn based on the data presented.

Reviewer #2: Yes

3. Has the statistical analysis been performed appropriately and rigorously?

Reviewer #2: Yes

4. Have the authors made all data underlying the findings in their manuscript fully available (please refer to the Data Availability Statement at the start of the manuscript PDF file)?

Reviewer #2: Yes

5. Is the manuscript presented in an intelligible fashion and written in standard English?

Reviewer #2: Yes

6. Review Comments to the Author

Reviewer #2: The authors have chosen not to implement the recommendations from the latest review into the manuscript, citing the closure of the study. Instead, they provide theoretical responses to the queries raised. Consequently, I leave it to the editor's discretion whether to proceed with the publication of the manuscript, taking into account the author's direct response to the comments.

2 ...Unfortunately, the study is now closed and it is not possible to conduct further data collection at this stage.

3… but we did not develop the app being evaluated and it was not designed to capture this data. We have already discussed this limitation in the Strengths, limitations, and future research section.

Upon review, I find that the comments have not been fully addressed. However, given the significant impact of the study, I defer to the editor's judgment regarding its suitability for publication. From my perspective, the responses provided only partially address the queries posed.

7. PLOS authors have the option to publish the peer review history of their article (what does this mean?). If published, this will include your full peer review and any attached files.

**Do you want your identity to be public for this peer review?** For information about this choice, including consent withdrawal, please see our Privacy Policy.

Reviewer #2: None
